# An Estimation of Consumer Spending on Sport in Lithuania, Its Division between Active and Passive Participation and the Effects of COVID-19

Themistocles Kokolakakis [1,*] and Vilma Čingienė [2,*]

1   Sport Industry Research Group, Sheffield Hallam University, Sheffield S1 1WB, UK
2   Faculty of Public Governance and Business, Mykolas Romeris University, LT-08303 Vilnius, Lithuania
*   Correspondence: t.kokolakakis@shu.ac.uk (T.K.); v.cingiene@mruni.eu (V.Č.)

**Abstract:** The economic importance of sport has been developing from several methodological origins. Most economic research into the sport economies develops indicators for gross value added (GVA), employment and consumer spending. A further elucidation of the benefits of the sport economy relates to well-being outcomes, either from sport participation or from sport spectating. The added value of this research is that it estimates sport consumer spending in Lithuania into two distinctive strands: active participation (e.g., participating in sport or fitness) and passive participation (e.g., attending sport events). The aim of this research is to link the consumer spending results to the GVA and employment results of the Sport Satellite Account and elucidate the main characteristics of the sport economy and how these are affected by the COVID-19 epidemic. Analysis of the scientific literature, a survey sample in Lithuania consisting of 3506 respondents who spent part of their household budget on sports activities (active or passive), and a specific method of analysis were developed and applied in this research. The research results show that consumer expenditure in sport is divided into passive and active with percentage shares of 17% and 83%, correspondingly. Sport consumption overall accounts for 2.2% of total consumption in Lithuania. Most elements of consumer spending in the passive and active categories are associated with welfare effects. The importance of the analytical framework is that we can explicitly differentiate among categories such as tourism for participation and tourism for spectating, which are associated with different well-being effects. The pandemic affected 53.7% of active sport consumption, a smaller percentage than in the case of passive participation (67.4%).

**Keywords:** economic importance of sport; household expenditure; sport participation; sport well-being

## 1. Introduction

In 2012, following the initiative of the Department of Physical Education and Sports of the government of the Republic of Lithuania and in collaboration with the National Statistics Office, the decision was taken to collect sport economic indicators to evaluate the economic importance of sport. The impetus for these processes increased because of the Lithuanian presidency of the EU council in the second half of 2013. One of the two selected topics proposed by the European Commission and agreed upon during the implementations of the EU Sport Policy was the economic impact of sport on the economy.

The main research outcome of the current work is an assessment of the structure of consumption in the sports sector based on a Lithuanian survey. Consumer-spending research acquired renewed importance because of the impact of the COVID-19 epidemic on the sport economy. All issues of assessing the sustainability of sport are literally filtered from an understanding of the existing consumer behaviour and how this is affected by the pandemic. For example, the restrictions on the operations of sport clubs and tourist units would directly affect the core of the sport economy, if one takes into account the

structure of most European sport economies as they appear in the pan-European sport satellite account [1].

The research into the economic importance of sport has been developing from several methodological origins. One is the overall spending on sport by consumers either for reasons of attending events or for reasons of personal participation [1]. Most economic research into sport economies develop indicators for gross value added (GVA), employment and consumer spending. A further elucidation of the benefits of the sport economy relates to well-being outcomes, either from sport participation or from sport spectating [2].

A sport satellite account has developed in several EU countries [1]. However, the tools for analysing the sport economy do not translate easily into sustainability outcomes, partly because there is no identification of the elements of sport consumption into participation and passive consumption (such as in spectating). The added value of this research is that it estimates sport consumer spending in Lithuania into two distinctive strands: active participation (e.g., participating in sport or fitness) and passive participation (e.g., attending sport events). Such a distinction can be used to develop a research tool link consumer research with sustainability and well-being research, where the active and passive categories are clearly identified.

The current study offers further added value to existing research by calculating consumer spending on sport in Lithuania for the first time. It separates the participation from the passive element of sport consumption through direct survey questions: each sport has been entered twice, once for active participation and another for passive participation. A further aim and added value of this research is to link the consumer spending results to the GVA and employment results of the sport satellite account (SSA), recently developed in Lithuania, and elucidate the main characteristics of the sport economy and how these are affected by the COVID-19 epidemic.

From this point of view, we do not put forward a formal hypothesis, as the effect of the COVID-19 pandemic on the sport sector is more than obvious; instead, we aim to estimate magnitudes and relationships relating to the consumer spending on sport and, through them, the effect of the pandemic.

## 2. Theoretical Background

The article by Ryan and Deci [3] defines the research of well-being as derived from two starting points: the hedonic approach, which focuses on happiness and defines well-being in terms of pleasure attainment and pain avoidance; and the eudemonic approach, which focuses on meaning and self-realization and defines well-being in terms of the degree to which a person is fully functioning. The well-being derived from sport is associated with both approaches; however, sport participation is mainly affected by the eudemonic approach. As such, it is easier to attach social characteristics to consumption and community engagement.

Of special importance are sport events, which have been recognised as mechanisms of increasing well-being [4] from the standpoint of spectating. The authors refer to research in both USA and Australia to establish the aforementioned effect. They conclude that people with strong team identification seem to be associated with greater well-being, most likely because team identification implies an element of community integration and emotional support. These people are also likely to feel a greater sense of pride. In addition, there is a trickle-down effect associated with sport events which brings participation benefits through increased sport membership. For example, according to the Weimar, Wicker and Prinz et al. [5] study, membership in sport clubs in Germany increases through home events, sport success, or the example of major athletes as role models. The link between sport success in events and sport membership is further underlined by the article of Castellanos-Garcia, Kokolakakis et al. [6]. All these effects, through increases in membership, convey increases in sport participation and well-being. They also underline the importance of separation between the active and passive elements in consumer behaviour, which is critical for the identification of the effects of COVID-19 on the sport economy.

This is further emphasised by the effects of sport on communities. According to Warner and Dixon [7], experiencing a strong sense of community is fundamental to one's overall life quality, well-being, and health. Thus, if we can find ways, through sport, for more individuals to feel strong social support at the group level, then it is possible to improve overall life quality and ensure that this growing need for community is met. By taking advantage of the inherent need to feel a sense of belonging to communities, sport organisations and clubs can offer the opportunity to their consumers and/or stakeholders to become a part of their community. Doing so would increase the engagement of their stakeholders, and, consequently, the commitment to the organization. Warner and Dixon highlight several research works that show that sport is one of the few remaining social institutions in which people regularly gather and experience community [8,9]. Such experience can be realised very strongly through active participation, but also through a 'passive' attachment to a club.

Attachment to clubs and participation effects are being associated with researched topics, such as mental and social well-being. They are also central concepts in policy tools aiming towards higher community and social integration. Using data from a sample of Australian Rules Football consumers, Doyle et al. [10] employ a positive psychological approach to examine the benefits of sport consumption (such as spectating) on individuals, in order to link sport to five domains of well-being: positive emotions, engagement, relationships, meaning, and accomplishment (PERMA). The study examines ways that sport consumption can reinforce the aforementioned well-being outcomes. Further, a review of 135 relevant spectator manuscripts published between 1990 and 2014 demonstrated that health (47%) and physical well-being (24%) are well-researched; however, domains of mental (20%) and social well-being (15%) receive much less attention [11].

Filo and Coghlan [12] explore the well-being effects associated with charity sport-event experiences. Such events provide charities with opportunities to promote the organization's mission, while providing participants with the opportunity to support its cause through participation. This research applies positive psychology to investigate well-being dimensions present in the event experience, similar to the PERMA framework of Doyle et al. [10]. Content analysis results indicate that all five domains of PERMA emerged as significant to varying degrees.

Sport participation and spectating consumption often involves travelling, sometimes in combination with vacations. From this point of view, it is important to highlight the impact of sport tourism on economic and well-being effects. Chen and Petrick [13] present a literature review of the effect of travelling on life satisfaction. The results revealed that the positive effects of travel experiences on perceived health and wellness have been demonstrated by multiple studies. These benefits have been found to gradually diminish after a vacation. It was also found that there is a lack of research demonstrating the positive effect of travel experiences on physical health. As before, the suppression of travelling by COVID-19 illustrates, conversely, the negative impact of the epidemic on well-being.

Another aspect of sport consumer behaviour is illustrated by Hedlund [14]. His article examines consumption patterns through membership and participation in sport-fan consumption communities. Sport consumers rarely attend sporting events alone. Instead, they participate in networks of like-minded fans and engage in collaborative and co-creation consumption activities. In sports, these networks are called sport-fan consumption communities. The results suggest that both membership and participation in such networks lead to increases in future intentions to attend games, purchases of the team's merchandise, and recommendations of the team's games to others. This result is fully consistent with the results of Inoue, Sato and Filo [4], who demonstrate the strong association of well-being and sports events. In the absence of such physical networks, during the pandemic, there is a need to substitute physical for virtual networks as much as possible in order to reproduce the aforementioned positive effects thereafter.

Kim Trail and Ko [15] examined the relationship between consumption in sport and the quality of association with the team in question. In line with other findings, they

found that the latter explained 56% of the variance in the intention to attend games, 75% of the intention to consume sport media, and 66% of the intention to purchase licensed merchandise. This research work reaffirms the strong relationship between passive sport consumption and the ties that people have with a sport club. This relationship is analysed in terms of trust, commitment, intimacy, self-connection, and reciprocity. Further, the participation benefits of active sports are well-identified. The social return on investment in sport [2] provides an analytical breakdown of benefits in terms of health, social and personal attainment. These insights justify the design of the current questionnaire into passive and active strands.

An important methodological insight of sport consumer spending is provided by the 2012 study of Preuss, Alfs and Ahlert [16] into the economic dimensions of sport consumption in Germany. Their study calculated sport consumption and also analysed its relations with numerous socio-demographic factors. As in the current study, it provided important information for the construction of a German SSA. The aforementioned authors concluded that consumer spending in sport in Germany was predominantly related to active sport participation (EUR 112.6 bn for the year 2010) and to a lesser extent to a passive interest (EUR 26 bn). To the extent that this could be generalised to a dominant characteristic of sport consumption, it would reinforce the well-being characteristics of sport.

Research by Downward and Rasciute [17] suggested that sport demand can be analysed relatively to the demand for other leisure, a structure suggested by micro-economic theory. This result underlines that sport consumption would change relatively slowly in line with leisure spending and that it would depend on general factors such as the availability of leisure time. In other words, some important determinants of sport demand are outside the strict parameters of sport and are affected by developments in the economy.

Wicker, Hallmann and Zhang [18] examined the consumption of sport generated by attending small sport events. They identified that people with higher income or of foreign nationality spent significantly more money at the events; event participants (athletes and coaches) had higher expenditures than spectators and volunteers; and event participants were more likely to revisit marathon events, but not the city, than spectators.

Honglan [19] showed that, in China, sport consumption was in a co-integrated relationship with disposable income. Further, disposable income is the Granger cause of actual sports consumption expenditures, but, on the contrary, actual sports consumption expenditures is not the Granger cause of disposable income. This result resonates the aforementioned result of Downward and Rasciute [17] on the relationship between sport consumption and leisure.

Elaborating on the analysis of Anderson et al. [20,21] and making it sport specific, we can state that, on one hand, service entities such as employees, sport clubs or sport enterprises and, on the other hand, consumer entities such as, individuals, families, or communities, provide the basis of the consumer behaviour analysis. Hence, much of the developments in sport relate directly or indirectly to consumer spending. For example, an increase in sport participation would increase more than proportionally spending on sport as new participants require new sportswear and equipment for their activity. However, the existing European research does not differentiate between active and passive elements. The current research endeavours to move the research in this direction. The remainder of the article explains the data issues around the sport consumption survey in Lithuania, presents the results and derives conclusions, including the effect on the sport sector of the COVID-19 pandemic in Lithuania.

## 3. Data Research Design and Methodology

For a contextual analysis of sport consumption in terms of demographic associations, a starting point is provided by Lera-López and Rapun-Gárate [22]. Their economic analysis provides many insights in terms of consumption and participation models.

The research design in the case of the SSA is determined by the detailed Vilnius definition of the sport industry. This defines the sport-related activities using detailed Statistical

Classification of Products by Activity (CPA) codes. These activities have been agreed at a Pan-European level and provide a consistent base for constructing SSAs [1]. The attraction of this definition is that it investigates sport in areas not traditionally associated with sport, such as financial services or sea transportation, providing a full and accurate picture of its economic importance. Each category in the Vilnius definition can be associated with GVA and employment but not necessarily with consumer spending. For example, sport goods manufacturing represents a flow to the retail and export sectors rather than directly to consumers. Further, some categories in the Vilnius definition would represent both passive and active consumption without any means of differentiating between them (e.g., spending on literature, merchandising, etc.). For this reason, two separate questionnaires were designed, one for consumer spending that relates to sport participation and one for passive consumption. Representative probability sampling was used to select respondents.

The goal of the survey was to analyse the annual expenditure of Lithuanian citizens on active or passive sports activities in 2014, use the data for the compilation of the Sport Satellite Account and reveal the national sport consumer's profile.

The survey employed two methods: CAPI (computer-assisted personal interview) and CAWI (computer-assisted web interview). A modified instrument, developed and applied in Germany [16], was used. The survey sample in Lithuania consisted of 3506 respondents, who spent part of their household budget on sports activities (active or passive), according to six socio-demographic criteria: age (15–75 years), sex, education, social status, place of residence, and average income per household. Calculations were made using the SPSS program version 19.0 (IBM, Armonk, NY, USA). Data about active consumer spending (24 categories) and passive consumer spending (12 categories) on 71 sports was collected. Active sport spending is associated with footwear, sportswear, sports and rental equipment, repairs, sports facilities, training sessions, capacity diagnostics, reading, computing, car expenditure for training or competition, public transport for training or competition, staying overnight, membership, insurance, medical services, medical supplies, sport camps, food and body care. Passive sport spending is associated with: fun items, tickets, car transport, public transport, accommodation, bars, TV, reading, betting, donations, passive memberships, and computers. Table 1 shows all activities that were surveyed. The list of activities is based on the German study of Preuss et al [16].

**Table 1.** List of all sports in the survey.

| 1. American Football | 19. Fencing | 37. Motorsports | 55. Acrobatics |
|---|---|---|---|
| 2. Badminton | 20. Fitness | 38. Nordic Walking | 56. Sport Fishing |
| 3. Ballet | 21. Football | 39. Pilates/Qi Gong/Tai Chi/Yoga | 57. Yachting/Pleasure Craft |
| 4. Baseball/Softball/Cricket | 22. Health Sports | 40. Cycling | 58. Squash |
| 5. Basketball | 23. Weightlifting | 41. Rasenkraftsport | 59. Surfing |
| 6. Beach Volleyball | 24. Paragliding | 42. Equestrian | 60. Dancing |
| 7. Alpine Climbing | 25. Golf | 43. Wrestling | 61. Diving |
| 8. Biathlon | 26. Gymnastics | 44. Skate Sports | 62. Tennis |
| 9. Billiard/Snooker | 27. Handball | 45. Rowing | 63. Table Tennis |
| 10. Bobsleigh/Luge | 28. Hockey | 46. Rugby | 64. Triathlon |
| 11. Bodybuilding | 29. Inline skating | 47. Chess | 65. Gymnastics |
| 12. Archery | 30. Martial Arts | 48. Shooting | 66. Ultimate Frisbee |
| 13. Bowling | 31.Canoeing/ Kayaking | 49. Swimming | 67. Volleyball |
| 14. Boxing | 32. Rock Climbing | 50. Aviation Sports | 68. Hiking |
| 15. Curling | 33. Running | 51. Sailing | 69. Waterball |
| 16. Ice Hockey | 34. Athletics | 52. Skateboarding | 70. Water Skiing |
| 17. Ice Skating | 35. Miniature Golf | 53. Skiing | 71. High Diving |
| 18. Parachuting | 36.Modern pentathlon | 54. Snowboarding | |

Source: Lithuanian Survey of Sport Consumer Spending; Preusss and Ahlert (2012) [16].

Each sport was associated with active and passive elements of consumption, in relation to sport participation. As a result, the following method of analysis was developed:

- Active and passive participants of the sport market were identified. The former mostly participate in sport through an activity, while the latter through spectating.

- Cost categories of active and passive sport participation were determined. A full list of passive elements of sport consumption are listed in Table 2, including using cars for going to events, spending money in bars during events, a part of TV spending associated with sport (as dictated by the hours of sport programming compared to the total), etc. Similarly, a full list of active elements of sport consumption is presented in Table 3, including sport camps, membership spending, car spending for going to training sessions, sport equipment, sportwear, etc. It is important to emphasise that some costs, such as use of cars or computers, are classified as passive or active sport consumption according to the purpose of use. If, for example, a car is used for going to events, it contributes to passive spending; however, if it is used for going to the gym, it contributes towards active sport expenditure.
- Using the survey results, the popularity of sports among active and passive sport consumers was measured. This was performed for each sport, asking directly if the participation is active or passive or both.
- Using the number of active and passive participants, as well as the corresponding active and passive consumption codes, the average expenditure on active and passive sport participation was calculated.
- The average expenditure on active sport participation was analysed by cost categories in the most popular sports.
- The total expenditure on active and passive sport participation was evaluated.

**Table 2.** 'Passive' sport consumption, Lithuania, 2014.

| Categories of Expenses | €m | % |
|---|---|---|
| Car for events | 19.14 | 16.7% |
| Bars | 19.06 | 16.7% |
| Tickets | 18.89 | 16.5% |
| Accommodation | 15.97 | 14.0% |
| TV spending | 12.71 | 11.1% |
| Computers | 7.34 | 6.4% |
| Fun items, retail | 5.41 | 4.7% |
| Donations | 4.18 | 3.7% |
| Public transport | 4.00 | 3.5% |
| Reading | 3.92 | 3.4% |
| Passive membership | 2.30 | 2.0% |
| Betting | 1.52 | 1.3% |
| TOTAL | 114.43 | 100% |

Source: Results of current research.

From this basis, together with the participation characteristics of adults in Lithuania [23], we derive for the first time the overall sport consumer spending in Lithuania.

The European Commission's Special Eurobarometer 472 on sport and Physical Activity [23] provides a snapshot of sport participation and motivations in the EU member states. In the case of Lithuania, 33% of the adult population exercise or play sport regularly or with some regularity compared with an average of 40% for the EU as a whole.

For the purposes of methodology, it is important to emphasise that the results of our survey elucidate the characteristics of sport participants but are not sufficient to establish the national dimensions of sport related consumer spending. To do that, the weights of sport participants suggested by the European Commission's Special Eurobarometer were used. For example, the average sport spending per participant derived through the survey is associated with a third of the Lithuanian adult population, following the national results of the Eurobarometer survey. From this point of view, the current results are as valid as the general classifications of the Eurobarometer survey: the Lithuanian survey here is both random and many times larger in participants.

The most important reasons for engaging in sports in Lithuania relate to 'improve your health' or 'to have fun'. Conversely, the most important reasons preventing sports participation are 'lack of free time' and 'lack of motivation and interest'.

**Table 3.** Active sport consumption, Lithuania, 2014.

| Categories of Expenses | €m | % |
|---|---|---|
| Sport camps | 69.43 | 12.8% |
| Membership | 69.25 | 12.8% |
| Sports equipment | 62.76 | 11.6% |
| Sportswear | 50.59 | 9.3% |
| Footwear | 46.69 | 8.6% |
| Car spending for training | 44.85 | 8.3% |
| Training sessions | 38.36 | 7.1% |
| Overnight staying | 19.75 | 3.6% |
| Sport facilities | 16.19 | 3.0% |
| Medical service | 16.05 | 3.0% |
| Rental of equipment | 15.87 | 2.9% |
| Insurance | 12.66 | 2.3% |
| Foodstuff | 10.22 | 1.9% |
| Medical supply | 9.64 | 1.8% |
| Public transport for training | 9.60 | 1.8% |
| Car spending for competition | 9.00 | 1.7% |
| Repairs of equipment | 8.98 | 1.7% |
| Car spending for competition abroad | 7.46 | 1.4% |
| Body care | 6.59 | 1.2% |
| Public transport for competition abroad | 4.38 | 0.8% |
| Computing | 4.35 | 0.8% |
| Reading | 4.02 | 0.7% |
| Capacity diagnostics | 2.53 | 0.5% |
| Public transport for competition | 2.38 | 0.4% |
| Total active | 541.58 | 100.0% |

Source: Results of current research.

## 4. Results

The survey produced the following headline results: Consumption of sport in Lithuania (2014) was estimated to be EUR 656 m. From this, around 83% (€541.6 m) was associated with active sport consumption, and the remainder (17%) with passive sport consumption.

The first step of the analysis is to identify the structures of active and passive sport participation in terms of specific sports. The questions of active and passive participation for each sport were asked in the survey and the results are depicted in Figures 1 and 2, below.

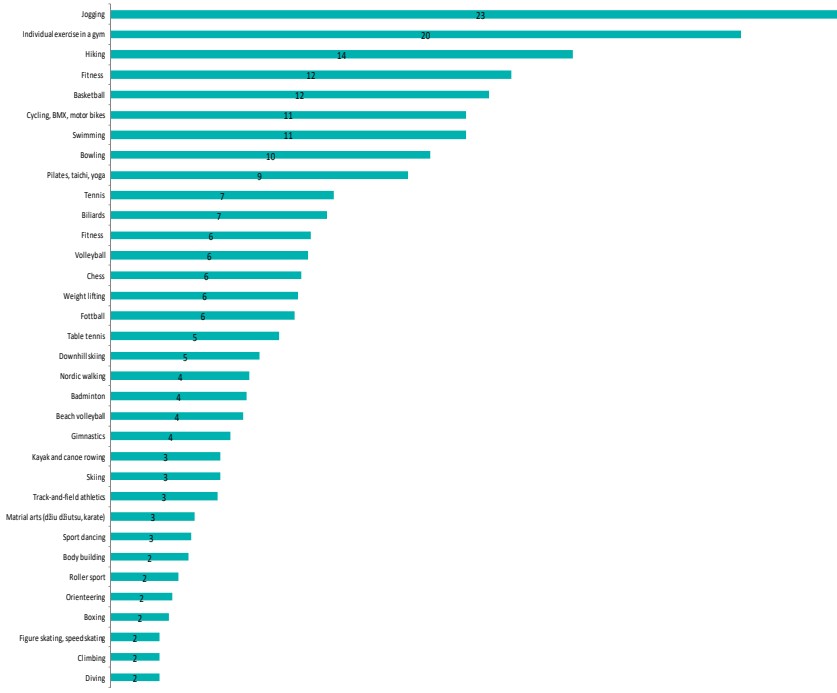

**Figure 1.** Sports in active participation, 2014. Source: Lithuanian Survey of Sport Consumer Spending.

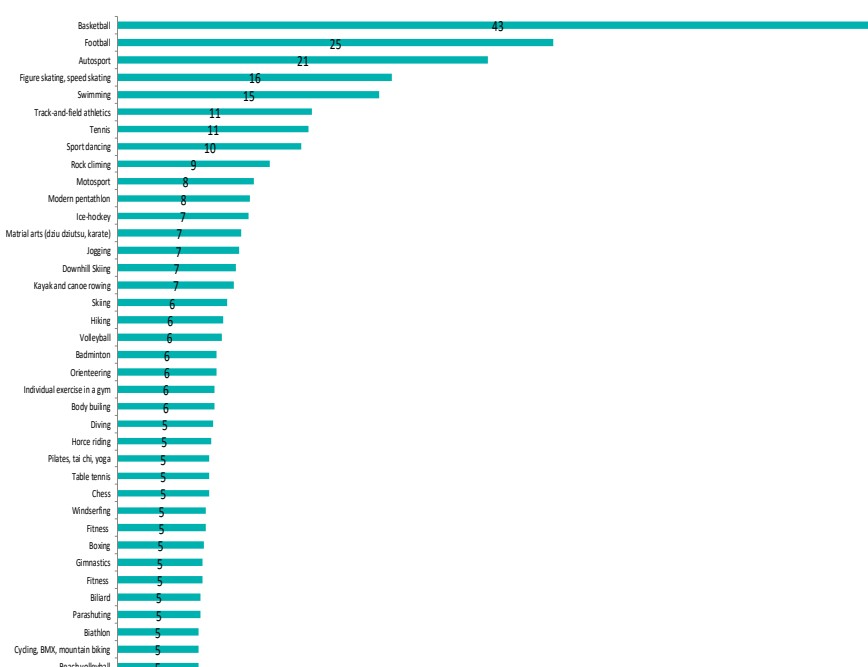

**Figure 2.** Sports in passive participation, 2014. Source: Lithuanian Survey of Sport Consumer Spending.

Figure 1, below, shows the importance of individual sports in the active-participation pattern within the sample of participants. The most important sport in terms of participation is jogging (practiced by 23% of participants), followed by gym (20%), hiking (14%), fitness (12%), and basketball (12%). This pattern emphasises sports that can be relatively cheap and easy to join (such as jogging and hiking), health and fitness following the expansion of operators throughout Europe, and basketball, which is a traditionally popular sport in Lithuania.

Figure 2 shows the pattern of sport under the passive categories. In this case we do not have activity-related participation rates but engagement for passive reasons, such as attendance at a basketball match.

The most important sports in terms of passive attendance are: basketball (attended by 43% of passive participants), followed by football (25%), auto sport (21%), figure skating (16%) and swimming (15%). Basketball appears among the most important sports both in active and passive participation.

The percentages presented in Figures 1 and 2 are very reliable when 'translated' to the Lithuanian population due to the large sample size of participants (3506). In each case, the margin of error at the 95% level is 0.7% each side of the estimate. For example, in the case of active participation in fitness (12% participation among general participants in the survey), there is a 95% probability that the national equivalent participation is between 11.3% and 12.7%.

Next, by using the number of participants in Figures 1 and 2, the cost codes in the survey and the Lithuanian national sport participation implied by the Eurobarometer Survey (as explained in the methodology section) we obtain the consumption patterns in Tables 2 and 3, which show the detailed results of passive and active sport expenditure, correspondingly.

According to Table 2, the most important expenditures for passive sport consumption are: car spending for watching sport (16.7%), bar spending (16.7%), tickets for sport events (16.5%) and sport-related accommodation, when spectating, (14%). The overall passive sport consumption amounts to EUR 114.4 m. The smallest categories are sport betting (1.3%) and passive membership (2.0%).

In a similar way, Table 3, indicates the more detailed active sport consumption in Lithuania for the year 2014.

According to Table 3, the most important categories of active sport spending are sport camps (12.8%), membership (12.8), sports equipment (11.6%) and sportswear (9.3%). The smallest categories are capacity diagnostics (0.5%) and public transport for competition (0.4%). Overall, active sport consumption amounts to EUR 541.6 m.

Finally, Table 4, below, indicates the major statistics of the sport economy in Lithuania in percentage terms.

**Table 4.** Basic sport economic indicators in Lithuania, 2014.

| Consumer Spending Distribution | % |
| --- | --- |
| Consumer spending—passive | 17% |
| Consumer spending—active | 83% |
| Consumer spending as % of total | 2.2% |
| Employment as % of total | 1.5% |
| GVA as % of total | 0.8% |

Source: Survey of Sport Consumer Spending in Lithuania, National Statistical Office of Lithuania, SIRC.

Firstly, consumer expenditure in sport is divided into passive (e.g., attendances) and active with percentage shares of 17% and 83%, correspondingly. Sport consumption overall accounts for 2.2% of total consumption. Similarly, according to Lithuania's National Statistical Office, sport employment and GVA correspond to 1.5% and 0.8% of overall GVA and employment.

## 5. Discussion of the Results

Most elements of consumer spending in the passive and active categories are associated with welfare effects. Some of the categories, such as car expenses or reading, are reproduced in both Tables 2 and 3. The importance of the analytical framework is that we can differentiate explicitly among categories such as tourism for participation and tourism for spectating which are associated with different well-being effects. As Chen and Petrick [13] have pointed out, a large number of welfare effects in sport are materialised though travelling and tourism (accommodation). This is expressed through car expenses, accommodation, and associated bar spending. An additional important effect is attending sport events as expressed by 'tickets' in passive consumption. Following the active and passive distributions, the tables can be used to allocate consumer spending among active and passive categories. For example, if the Lithuanian consumer behaviour is a valid indicator of the general European consumer pattern, then membership for active pursuits should be about four times greater than membership for passive participation. Such information can be used by National Statistic Offices when a distribution between active and passive consumption is required (such as in the UK, [24]).

The analysis of the survey results revealed relations between traditional sports and the forms of physical activity. Basketball, swimming and tennis remain the most popular sports and receive the biggest part of consumer expenditure. Individual sports and different forms of physical activity, such as running, exercising in fitness centres, bowling, pilates, and wellness sport prevail in the active sport participation market. A higher concentration of traditional sports is observed in the passive sport participation market. Presumably, traditional sports have gained interest among the Lithuanian population mainly as a spectacle. Nevertheless, passive observers of sports are potential consumer groups that may become active participants in sports if appropriate motivation measures and methods, family and community initiatives are applied. Active participation is dominated by physical activities such as jogging and health and fitness, which are likely to survive during times of pandemic. From this point of view, the sustainability of sport during a crisis could be maintained through active participation.

The key sport indicators of Table 4 show that the Lithuanian sport economy generates in percentage terms more employment (1.5%) than GVA (0.8%). This is a common pattern that is reproduced throughout the EU [1] and implies that sport is an efficient generator of employment. For example, in a period of recession, when increases in employment

are usually an important policy goal, investing EUR 1 m in the sport industry would generate more employment than average. Note, however, that the percentage share of sport consumption surpasses both the percentages of GVA and employment. This is likely to be associated with a widespread consumption pattern, as in every other EU country, and, at the same time, a more restricted sport-industry manufacturing and service base due to the size of the national economy. Conversely, this means that a targeted development of sport services and manufacturing can potentially be absorbed by the domestic market, increasing the effectiveness of generating sport employment even further.

## 6. Conclusions and Further Research with Reference to the Effect of COVID-19

This research reinforces the results of Preuss et al. [16] in the case of Germany, where the vast majority of consumer spending and well-being effects are in the active sport category (more than 80%). These are mainly expressed as spending on sport camps, membership for participation, sport equipment, footwear and sportswear.

Further, this research represents the first time that the consumer spending of sport for Lithuania is estimated using the active/passive methodological framework. To take the methodology further, we can take into account sport economic research of consumption such as by Scheerder et al. [25]. These authors examined consumer expenditure on sport clothing and footwear. They found that that the decision to spend money on sport clothing and shoes is mainly determined by sport-related lifestyle characteristics (sport participation and sport spectating), confirming the emerging importance of lifestyle in understanding the decision to consume material goods. Hence, the article raises the suggestion that sport elements such as footwear and sportswear may be consumed for passive reasons. In the UK experience [24], this is more likely to be important in the case of footwear. A lot of young people buy sport shoes as items of fashion rather than participation. In this case, this element of sport consumer spending will not be classified at all under the passive or active sport categories. A robust survey within the population as a whole would be necessary to evaluate this additional sport spending dimension.

Finally, the consumer results provide a tool for assessing the impact of COVID-19 on the sport market. Table 5, below, shows the effect of the pandemic during the lockdown of clubs, gyms, and accommodation.

**Table 5.** Effect of COVID-19 on the sport economy.

| Categories of Expenses | % |
|---|---|
| **Passive sport consumption** | |
| Car for events | 16.7% |
| Bars | 16.7% |
| Tickets | 16.5% |
| Accommodation | 14.0% |
| Public transport | 3.5% |
| Total % of Passive consumption | 67.4% |
| **Active sport consumption** | |
| Sport camps | 12.8% |
| Membership | 12.8% |
| Car spending for training | 8.3% |
| Training sessions | 7.1% |
| Overnight staying | 3.6% |
| Sport facilities | 3.0% |
| Public transport for training | 1.8% |
| Car spending for competition | 1.7% |
| Car spending for competition abroad | 1.4% |
| Public transport for competition abroad | 0.8% |
| Public transport for competition | 0.4% |
| Total % of Active consumption | 53.7% |

Source: Results of current research.

The first part of Table 5 shows the elements of passive sport spending that are affected by the pandemic. These include items such as car for events, bars, tickets, accommodation, and public transport. The table assumes cancelation of sport events and restrictions in tourism accommodation. It shows that COVID-19 would significantly affect 67.4% of the passive element of sport consumption.

The second part of Table 5 examines the effect of COVID-19 on active sport consumption. It includes items such as sport camps, membership (assuming no membership is charged during lockdown), car spending for training, training, etc. Other items associated with sport activities, such as sport clothing and equipment, are not affected as they can be bought online, and sport participation does not necessarily decline in all sports during the pandemic. Overall, the pandemic affected 53.7% of active sport consumption, a smaller percentage than in the case of passive participation. This inequality of the effects reflects two sport characteristics in Lithuania: firstly, that passive consumption is largely based on attending events and, secondly, that some sport activities such as jogging, hiking, and cycling are not affected by the pandemic, while others such as fitness can take place at home.

The results suggest that, at a grassroots level, the pandemic would have a devastating effect on club finances. All the major revenue streams associated with membership, attendance of events and accommodation or food services would decline dramatically, and the government would need to intervene to support the club network in order to maintain the current infrastructure. However, given its relative economic size, it is even more important for the national economy to prevent a fall in active consumer spending, which is linked to the sport participation rates. To perform some activities, such as jogging and cycling, participation must be encouraged in a safe way while the existing physical networks of participants (e.g., around a fitness centre) must be substituted by online networks (e.g., for fitness classes at home). Increasing sport participation can also balance some negative well-being effects from the cancelation of events. Well-being may reduce directly [4,10] or indirectly through declining membership (by reversing the trickle-down effect presented in Weimar, Wicker et al. [5].

Further, if we take into account the research by Warner and Dixon [7], the isolation of the pandemic would decrease the sense of community that can be cultivated through sport. To counteract this decline, sport agents must develop online networks for participation requirements and to address community problems.

Overall, the results provide implications for event managers to enhance the event experience, and bolster appeals for external funding in cases of COVID-19 epidemics, as well as a basis for further investigation of well-being and sport events in public health policies. In the above, one should apply the caveat that the consumption dataset is based on 2014 and, therefore, may be limited in terms of 'new' sports; however, the general relationships uncovered should still be valid in the light of how little sports economies change year to year throughout the EU.

**Author Contributions:** Conceptualization, T.K. and V.Č.; methodology, T.K.; software, T.K.; validation, T.K. and V.Č.; formal analysis, T.K.; investigation, V.Č.; resources, V.Č.; data curation, T.K.; writing—original draft preparation, T.K. and V.Č.; writing—review and editing, T.K. and V.Č.; visualization, V.Č.; supervision, T.K.; project administration, V.Č.; funding acquisition, V.Č. All authors have read and agreed to the published version of the manuscript.

**Funding:** The research was funded by Lithuania Physical Education and Sport Foundation (grant KSTP-12).

**Institutional Review Board Statement:** Not applicable.

**Informed Consent Statement:** Not applicable.

**Data Availability Statement:** Primary data in this study are based on the presented questionnaire.

**Conflicts of Interest:** The authors declare no conflict of interest.

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
