# Peer review of "An Estimation of Consumer Spending on Sport in Lithuania, Its Division between Active and Passive Participation and the Effects of COVID-19"

_sustainability, doi:10.3390/su141912261_

Round 1
Reviewer 1 Report
1. Abstract - please, add or specify keywords
2. Table 1, 2, 3, 5. Figure 1, 2 - please, add Source
3 …..Ry.an and Deci (2001)…….- not sighted in the References
4. References - Hedlund, D., P. (2014) …. Used twice
Author Response
Dear Reviewer 1
Thank you for your comments. Please see attached our answers to all Reviewers.
Your comments have really helped the paper.
Best wishes

Reviewer 2 Report
The article is actual and relevant. Sport has a multifaceted impact on the economy and on society as a whole. The authors of the article carried out an extensive review, which notes the multilateral influence of sports activity on the functioning of society. The economic effects of the functioning of the sports sector are noted. Social and psychological aspects are also highlighted. We should agree with the authors that the structure of consumption in the sports sector has not been sufficiently studied in the scientific literature. In addition, the Vilnius definition of sport does not affect the division of sports consumption into active and passive. Such a distribution will certainly provide additional valuable information about the structure of consumption in the field of sports and will clarify the impact of the sports sector on the economy.
The main result of this work should be called an assessment of the structure of consumption in the sports sector based on Lithuanian empirical data.
There are a number of comments on the presented work, as well as questions requiring clarification.
First, it is necessary to clarify why the authors use the term "motivation" to denote active and passive use in the sports field.
Secondly, the results obtained by the authors should be considered new: the authors clarify the structure of consumption in the sports sector according to Lithuanian data. But it is not clear from the text of the work what is its scientific novelty. In particular, the authors do not define new characteristics and relationships of the phenomena and processes under study, do not formulate hypotheses for further testing. Thus, the work is applied in nature. In it, on the basis of a survey, an assessment was made of the significance of some statistical indicators in the field of sports using the example of Lithuania.
Thirdly, the generalization of statistical calculations for the whole country according to the sample survey data used in the presented work is a well-known and promising tool. However, the scientific validity of its use is determined by a number of conditions: the sample must be representative; to generalize the dependencies and patterns identified during the survey, they must be strictly mathematically proven on the basis of, for example, regression analysis. The above questions are not covered in the article. As a consequence, there is no rigorous justification for using survey data to draw conclusions across the entire national economy.
In general, the conducted studies can become the basis for clarifying the system of statistical indicators in the field of sports both in Lithuania and in the countries of the European Union as a whole. However, obtaining reliable data on the structure of active and passive use in the field of sports will require significant efforts on the part of the statistical authorities, which may lead to a refusal to properly refine the structure of statistical indicators in the field of sports.
A few more local remarks should be made:
1. Table 1 lists the sports (kinds of sports activity). Obviously, this list is formed on the basis of a well-known statistical classification in the field of sports. And it would be enough to indicate the source of statistical information in which this list of sports is given, and not to present it in the text of the work in the form of a cumbersome table.
2. Given that the study is of an applied nature, it should be noted that the use of data for 2014 in the study reduces the relevance and value of the results obtained.
3. The assessment of the impact of the COVID-19 pandemic on active and passive consumption in the sports sector is interesting, however, it is private in nature and has no scientific value in itself.
Recommendations to authors:
1. It is necessary to
1) formulate and substantiate the provisions of scientific novelty, the scientific contribution of the work.
2) bring the literature review in line with the topic of the work: presently the review mainly affects the socio-psychological aspects of the impact of sports on its participants.
3) strictly mathematically (statistically) prove the characteristics and parameters of the relationships identified during the survey, present the results in the text of the paper.
2. Besides, from my subjective point of view, the work would have benefited if the authors had focused only on the topic of the article, devoted to the structure of active and passive consumption in the field of sports and its impact on the economy. The issues of the COVID-19 pandemic and the socio-psychological effects of sports activity deserve separate consideration.
Author Response
Dear Reviewer 2
Thank you for your comments. Please find attached our reply to all comments.
Your have helped significantly in improving the article.
Best wishes

Reviewer 3 Report
First of all, I am grateful for the opportunity to review this manuscript. In general terms, the subject matter is interesting, it is in line with the contents of the Journal and the objectives are well presented. I consider that the references should be updated, in addition to the fact that they should be presented according to the standards of the Journal (they are not well presented) and the Discussion section should be improved, since it is not very deep in terms of the ideas presented. It should also improve the practical implications of the study and its novelty in terms of research.
Best wishes to the authors.
Author Response
Dear Reviewer 3
Thank you for your comments. Please find attached detailed answers to all Reviewers. You have helped a lot to improve the article.
Best wishes.
